Immuno-oncologic profiling by stage-dependent transcriptome and proteome analyses of spontaneously regressing canine cutaneous histiocytoma

Loriani Fard Alina K. 1
http://orcid.org/0009-0002-0279-7482 Haake Alexander 1
http://orcid.org/0000-0002-8056-5065 Jovanovic Vladimir 2 3
Andreotti Sandro 2
Hummel Michael 4
http://orcid.org/0000-0002-1998-4033 Hempel Benjamin-Florian 5
http://orcid.org/0000-0002-4502-0393 Gruber Achim D. 1 achim.gruber@fu-berlin.de
1 Department of Veterinary Medicine, Freie Universität Berlin, Institute of Veterinary Pathology , Berlin , Germany
2 Department of Mathematics and Computer Science, Freie Universität Berlin, Bioinformatics Solution Center , Berlin , Germany
3 Institute for Biology, Freie Universität Berlin, Human Biology and Primate Evolution , Berlin , Germany
4 Charité University Medicine, Institute of Pathology , Berlin , Germany
5 Department of Veterinary Medicine, Freie Universität Berlin, Veterinary Center for Resistance Research , Berlin , Germany
Felley-Bosco Emanuela
Electronic publication date: 2024 Nov 26
Publication date: 2024
Volume: 12
Electronic Location ID: e18444
Received 2024 May 20; Accepted 2024 Oct 11
Copyright: © 2024 Loriani Fard et al.
Copyright year: 2024
Copyright holder: Loriani Fard et al.
License: This is an open access article distributed under the terms of the Creative Commons Attribution License, which permits unrestricted use, distribution, reproduction and adaptation in any medium and for any purpose provided that it is properly attributed. For attribution, the original author(s), title, publication source (PeerJ) and either DOI or URL of the article must be cited.
License URL: https://creativecommons.org/licenses/by/4.0/

Keywords: 3′ RNA-seq, Canine IO panel, Canine cutaneous histiocytoma, Co-stimulatory molecules, Pathology, Formalin-fixed, Paraffin-embedded, FFPE, Canine, Immuno-oncology, Tumor regression

Funding: Förderlinie TEAMS Mittel zur Förderung der Vernetzung zwischen Wissenschaftler/-innen der Freien Universität Berlin (TEAMS funding line Funds to promote networking between Scientists at the Freie Universität Berlin) This study was supported by “Förderlinie TEAMS Mittel zur Förderung der Vernetzung zwischen Wissenschaftler/-innen der Freien Universität Berlin” (TEAMS funding line Funds to promote networking between Scientists at the Freie Universität Berlin). There was no additional external funding received for this study. The funders had no role in study design, data collection and analysis, decision to publish, or preparation of the manuscript.

==============================
Canine cutaneous histiocytoma (CCH) is a tumor that originates from dermal Langerhans cells and affects particularly young dogs. The common spontaneous regression of CCH makes it an interesting model in comparative oncology research. Previous studies have indicated that anti-tumor immune responses may be involved, but details remain speculative to date. Here, we asked which specific immuno-oncological dynamics underlie spontaneous regression of CCH on mRNA and protein levels. QuantSeq 3′ mRNA sequencing with functional over-representation analysis and an nCounter RNA hybridization assay were employed on 21 formalin-fixed, paraffin-embedded CCH samples representing three different tumor stages (dataset information: GSE261387—Immuno-Oncologic Profiling by Stage-Dependent Transcriptome and Proteome Analyses of Spontaneously Regressing Canine Cutaneous Histiocytoma—OmicsDI). Nine additional samples were subjected to matrix-assisted laser desorption/ionization mass spectrometry imaging (MALDI-MSI). Surprisingly, only minor stage-specific differences were found. When we investigated expression of B7 family ligands and CD28 family receptors holding co-stimulatory and -inhibitory functions, respectively, we found a higher abundance of CD80, CD86, CTLA4 and CD28, which may trigger a balanced activation of lymphocyte-mediated immune responses. CD80 and CD86 expressing cells were further quantified by in situ hybridization and compared with data from three cases of canine histiocytic sarcoma (HS), a malignant tumor variant originating from antigen-presenting interstitial dendritic cells. A stage-specific increase of CD80 expressing cells was recorded in CCH from the tumor bottom to the top, while CD86 was continuously and homogenously expressed at high levels. Overall expression of CD80 in CCH was similar to that in HS (73.3 ± 37.4% vs 62.1 ± 46.4%), while significantly more CD86 expressing tumor cells were found in CCH (94.7 ± 10.3%) when compared to HS (57.6 ± 11.0%). Our data suggest that major immuno-oncological pathways are not regulated during regression of CCH on the mRNA or protein levels as detectable by the methods used. Instead, our data provide further evidence supporting previous hypotheses towards a role of immune stimulatory B7 family ligands and CD28 family receptors in the regression of CCH.

Introduction

Immunological approaches to combat cancer are among the most prioritized fields of research both in human and veterinary oncology. Specifically, approaches are favored that engage the host immune system to successfully attack and eliminate tumor cells (Xu et al., 2023). A sound understanding of the mechanisms involved is therefore among the most crucial prerequisites: Specifically, how can host immune cells recognize and kill tumor cells, and how can they be trained to do so successfully?

Canine cutaneous histiocytoma (CCH) is a common skin tumor of young dogs most likely originating from a subset of dendritic cells (DC) of the skin, termed Langerhans cells (Marchal et al., 1995; Moore et al., 1996). In addition to its rapid growth and high mitotic activity (Glick, Holscher & Campbell, 1976), CCH is characterized by a stereotypical course, which in the vast majority of cases results in spontaneous regression (Moore et al., 1996; Pires et al., 2009). Moreover, there is both a time-dependence of regression and a spatial gradient involving a bottom-heavy immune reaction that spreads throughout the tumor (Cockerell & Slauson, 1979). This unique feature makes CCH an interesting natural animal model for spontaneous tumor regression, likely involving anti-tumor immune mechanisms.

Based on stereotypical patterns of structural changes and lymphocytic infiltrates, the tumor was classified into four morphological stages (Cockerell & Slauson, 1979). From stage 1 to 4, the number of infiltrating lymphocytes increases, starting at the basolateral periphery of the tumor and spreading across the center until, in the final stage 4, the number of inflammatory cells by far exceeds that of the tumor cells. After initial influx of CD4+ T cells, an increase in pro-inflammatory cytokines, namely IL-2, IL-12, TNF-α and IFN-γ, as well as nitric oxide synthase an indicator of macrophage activation, has been observed, followed by an influx of CD8+ T cells (Kaim et al., 2006). These observations suggest that a classical anti-tumor immune response may underlie the regression of CCH, which is mainly based on CD8+ lymphocyte-mediated anti-tumor effects (van der Leun, Thommen & Schumacher, 2020). However, it is still unclear by which molecular and signaling mechanisms this anti-tumor immune response is triggered.

Langerhans cells, or progenitors thereof, are thought to be the cells of origin for CCH (Marchal et al., 1995; Moore et al., 1996). As a special type of DC, they serve as major regulators between the innate and adaptive immune systems and thus represent a key component of the dermal immune defense. In particular, they play a crucial role in regulating types of T cell immune responses, specifically in anti-tumor immunity (Diamond et al., 2011; Fuertes et al., 2011). In their naïve state, their main task is to recognize antigens which upon activation are presented via major histocompatibility complexes (MHC) classes I and II to lymphocytes in tributary lymph nodes (Banchereau & Steinman, 1998; Steinman, 2012; Steinman & Banchereau, 2007). Two synergistic cellular interactions mediated by cytokines are mandatory for DC-mediated lymphocyte activation and proliferation (Bretscher, 1999; Jenkins & Schwartz, 1987), also referred to as the “two signal model” (Chen & Flies, 2013; Sharpe & Freeman, 2002). The first signal consists of T cell receptor (TCR) binding to MHC I or II antigen-complexes on the DC while the second refers to binding of co-stimulatory immunoreceptors on lymphocytes to their ligands expressed on antigen-presenting cells (APCs), including DCs (Pan et al., 2023). Specifically, the CD28 family receptors expressed by lymphocytes and their B7 family ligands on APCs acting as immune-checkpoint molecules are crucial for the activation or inhibition of the T cell immune responses. The former group consists of CD28, cytotoxic T lymphocyte-associated protein 4 (CTLA4), programmed cell death protein 1 (PD-1) and inducible co-stimulator (ICOS). On the other hand, the B7 family ligands contain CD80, CD86, programmed cell death protein 1 ligand 1 (PD-L1) and 2 (PD-L2), inducible co-stimulator-ligand (L-ICOS), B7 Homolog 3 (B7-H3, CD276) and V-set domain-containing T-cell activation inhibitor 1 (B7-H4, VTVN1).

In terms of expression by cell type and their activation, the interaction of CD80 and CD86 on APCs with CD28 on lymphocytes induces activation and proliferation of T lymphocytes (Chambers & Allison, 1997; Lenschow, Walunas & Bluestone, 1996; Lucas et al., 1995; Shahinian et al., 1993). In contrast binding of CD80 and CD86 to CTLA4 inhibits a T cell mediated immune response (Tivol et al., 1995; Walunas et al., 1994; Waterhouse et al., 1995). The PD-1 receptor serves as an important inhibitory immune checkpoint which binds to its ligands PD-L1 and 2 on APCs (Ghiotto et al., 2010; McDermott & Atkins, 2013; Ohaegbulam et al., 2015), with immunosuppression being mainly induced by binding of PD-1 to PD-L1 (Brahmer et al., 2010). ICOS on lymphocytes delivers a positive co-stimulatory signal via binding to its ligand L-ICOS on APCs resulting in the activation and differentiation of T lymphocytes (Hutloff et al., 1999). B7-H3 and B7-H4 on APCs mainly possess inhibitory functions, suppressing T cell activation and proliferation via still unknown receptors (Hofmeyer, Ray & Zang, 2008; Prasad et al., 2004; Sica et al., 2003; Zang et al., 2007).

It has been postulated that CCH tumor cells, as they originate from Langerhans cells, increasingly exhibit the phenotype of mature APCs over time, from bottom to top of the tumor, which enables them to trigger an effective immune response (Baines et al., 2007; Diehl & Hansmann, 2024; Pires et al., 2009; Pires et al., 2013b). This notion was supported by the observation that CCH cells are potent stimulators in the allogenic mixed leucocyte reaction (Baines et al., 2007). In addition, decreased expression of E-cadherin (Pires et al., 2009), a marker of immature Langerhans cells, and increased levels of CD206, a marker of activated dendritic cells, were found over time. This picture was further supported by increased expression of Iba-1 during tumor regression (Belluco et al., 2020), a marker of antigen-presenting cells. Finally, expression of MHC II-shifted from the cytosol to the tumor cell surface (Kipar et al., 1998) and progressed from basolateral to apical expression within the tumor, corresponding to the distribution of infiltrating lymphocytes (Pires et al., 2013b).

To this day, it has been unknown whether other proteins in addition to MHC II molecules or unrelated pathways may be involved in the immune response that triggers interaction between lymphocytes and tumor cells, which found recently renewed interest (Diehl & Hansmann, 2024). This includes whether CCH regression involves increased expression of CD80 and CD86, two well established markers of DC maturation, by tumor cells (Banchereau et al., 2000; Palucka & Banchereau, 2012).

In this study, we asked which specific immuno-oncological dynamics may underlie regression of CCH on mRNA and protein levels. In a primary hypothesis-generating approach, we tested for stage-dependent differences of tumor cells and lymphocytes, on the transcriptome and proteome levels. To this end, we analyzed archival samples from spontaneous CCH tumors after applying rigorous inclusion and exclusion criteria. To account for limitations of formalin-fixed paraffin-embedded (FFPE) material in terms of RNA fragmentation due to chemical modification caused by formaldehyde fixation (Masuda et al., 1999; von Ahlfen et al., 2007; Werner et al., 2000), we employed the QuantSeq 3′ method at whole transcriptome level and the nCounter Canine IO Panel for quantifying expression of immuno-oncologically relevant genes. Both methods allow the identification of short fragments of RNA and therefore are well suited for investigations employing FFPE material (Jang et al., 2021; Manjunath et al., 2022). Furthermore MALDI-MSI was utilized for global proteome analyses with spatial resolution. The resulting data were used to assess classical pathways of pro- and anti-tumor responses. Based on the hypothesis that CCH tumor cells take on the phenotype of mature APCs over time and thus trigger anti-tumor cell responses (Belluco et al., 2020; Diehl & Hansmann, 2024; Moore, 2016), we further asked whether they express relevant co-stimulatory molecules and quantified the expression of CD80 and CD86 in a tumor stage-dependent and spatial manner via in situ hybridization. Finally, we compared expression of these genes in CCH with their expression in canine histiocytic sarcoma (HS), a malignant tumor variant originating from APCs that does not undergo regression. With this approach, we tested the hypothesis that the expression of CD80 and CD86 may play a role in the immune recognition of tumors that originate from APCs.

Materials and Methods

Selection of tissue samples

Tissue samples of a total of 300 formalin-fixed, paraffin-embedded (FFPE) CCH tumors from privately owned dogs were obtained from the archive of the routine diagnostic biopsy service at the Institute of Veterinary Pathology, Freie Universität Berlin, spanning the years from 2010 to 2022. Tumors had been surgically excised for curative or diagnostic purposes only. An assessment by the responsible authority certifies that the retrospective use of such archival tissue at this institute does not qualify as an animal experiment under § 7 of the German animal protection act (decision StN 010/23 of the State Office for Health and Social Affairs, Berlin). Dog owners had given consent to the use of the tissues for research purposes by agreeing to the general terms and conditions of the diagnostic service. Selection of tumors was based on the unequivocal histopathological diagnosis on hematoxylin and eosin stained microscopical slides by a European board-certified veterinary pathologist according to accepted criteria (Mauldin & Peters-Kennedy, 2016). These distinguishing criteria include the following features: round to oval tumor cells with moderate to abundant, eosinophilic, slightly foamy cytoplasm and a centrally located single, round to oval, indented or convoluted nucleus with vesicular chromatin. In addition, there is a stage-dependent lymphocytic infiltration that starts at the periphery of the tumor and progresses to the center and top. Tumors were then subjected to a second round of review by two independent veterinary pathologists (specialists and European board-certified) to rule out possible misdiagnoses. Only tumors that could be clearly diagnosed on the basis of their microscopic patterns were selected (Mauldin & Peters-Kennedy, 2016). The validity of the tumor diagnosis was later confirmed on the transcriptome level (See Results and Fig. S1). All CCHs were staged based on the presence and distribution of lymphocytes in the tumor (Cockerell & Slauson, 1979). Briefly, lymphocytes are absent from stage 1, moderately infiltrate the basolateral margin and center of the tumor in stage 2, or infiltrate and extend to the tumor center and form follicle-like structures in stage 3. In stage 4, lymphocytes outnumber tumor cells, necrotic areas are common and regression is discernible histologically. In this study, only stages 1, 2, and 3 were investigated. Stage 4 was excluded because of heavy tumor heterogeneity and large areas of tumor necrosis. Following our inclusion criteria (at least 1 cm size in the smallest dimension, absence of hemorrhage, hair follicles, deep purulent inflammation or other contaminating factors) a final number of 59 CCH tumors were included in this study. Canine splenic non-hemophagocytic histiocytic sarcoma (HS) samples were selected from the same departmental archive of diagnostic tissue specimens and processed accordingly (for further sample information see Table S1).

Preparation of tissue samples

To obtain meaningful transcriptome results from tissue extracts, large tumor areas were selected for preferably homogeneous cell populations with as little as possible non-tumorous components. Tumor-adjacent tissues and ulcerated tumor surfaces were removed from all samples. Stage 2 tumors per definition contained two separate, tumor-cell rich or lymphocyte-rich areas which were processed separately using a tissue array. To this end, areas of stage 2 tumors were selected under the microscope with at least 90% tumor cells (referred to as sample “group 2a” in the following) and areas with at least 60% lymphocytes (referred to as sample “group 2b” in the following). These areas were detached from the paraffin blocks using biopsy punches of 2, 3 or 4 mm in diameter and inserted into new paraffin blocks. The tissue transferred from a single block was sufficient for tumor cell sample group 2a in each case to create a new paraffin block with sufficient material from a single tumor for the subsequent RNA isolation. For the lymphocyte-enriched samples in group 2b, explants from three to four CCH were combined on a single array to obtain tissue arrays with sufficient material.

RNA extraction from FFPE samples and RNA quality control

For all steps prior to RNA extraction, surfaces and instruments were treated with RNase AWAY (Thermo Fisher Scientific Inc., Waltham, MA, USA). Five 10 μm FFPE scrolls per sample were prepared in sterile centrifuge tubes and shipped to Lexogen GmbH Services (Vienna, Austria) in sterile centrifuge tubes on dry ice. Total RNA was extracted utilizing the PureLink FFPE Total RNA Isolation Kit (Thermo Fisher Scientific Inc., Waltham, MA, USA) according to the manufacturer’s guidelines. Total RNA concentrations were measured with a NanoDrop 2000c spectrophotometer (Thermo Fisher Scientific Inc., Waltham, MA, USA) and quality was determined using an Agilent 5200 Fragment Analyzer (Agilent Technologies, Inc., Santa Clara, CA, USA) employing the DNF-471F33—SS Total RNA 15 nt—FFPE Illumina DV200 method mode. Only samples with a total RNA quality number (RQN) (Schroeder et al., 2006) of >4.1 and DV200 (percentage of RNA fragments over 200 nucleotides (nt) in length) (Matsubara et al., 2020) of >64.5% were processed. Data on RNA quality are provided in the Supplemental Information (Table S1).

3′ RNA sequencing (RNA-seq)

DNase I treated total RNA (375 ng) was sequenced (n = 5 or n = 6 in case of sample group 2b) using the QuantSeq 3′ mRNA-Seq FWD Library Preparation Kit (Lexogen, Vienna, Austria) according to the manufacturer’s guidelines (user guide 015UG009V0251) at Lexogen Services (Lexogen GmbH, Vienna, Austria) using the low-quality RNA protocol. Quality of the libraries was determined with an Agilent 5300 Fragment Analyzer (DNF-474-33—HS NGS Fragment 1–6,000 bp method mode). The samples were pooled in equimolar ratios. The library pool was quantified using a Qubit dsDNA HS assay kit (Thermo Fisher Scientific Inc., Waltham, MA, USA) and sequenced utilizing an Illumina NextSeq 500 system with a SR75 High Output Kit at Lexogen Services.

The FASTQ sequencing files were first preprocessed (adapter trimmed, filtered) using Cutadapt (Martin, 2011) and subsequently aligned to the NCBI Reference Sequence (RefSeq) assembly for the dog (Canis lupus familiaris) UU_Cfam_GSD_1.0 (NCBI RefSeq assembly GCF_011100685.1) with the STAR aligner (Dobin et al., 2013). Differential gene expression (DGE) analysis for the comparisons of group 2a vs group 1 and group 3 vs group 2b was performed with R using DESeq2 (Love, Huber & Anders, 2014). Significance thresholds were drawn at a log2(fold change) (log2FC) of ≤−1 or ≥1 and an adjusted p-value (padj) of ≤0.05 (McDermaid et al., 2019; Rapaport et al., 2013; Reimand et al., 2019).

In order to assess the reliability of the transcriptome data and the initial selection of CCH tumor samples, the expression levels of tumor markers of different canine round cell tumors (Paździor-Czapula et al., 2015) were analyzed using the normalized counts from the QuantSeq 3′ analysis (see Fig. S1).

Functional enrichment analysis

A functional enrichment analysis was carried out using g:Profiler (Reimand et al., 2016). Significantly differentially expressed genes from the comparison of sample group 2a vs 1 and sample group 3 vs 2b and specific peptides measured via MALDI-MSI (see below) were fed into the functional profiling tool g:GOSt. Default settings were retained under “options”. Subsequently, the calculated Gene Ontology (GO) (Ashburner et al., 2000; The Gene Ontology et al., 2023) terms from the GO categories molecular function (MF), cellular component (CC) and biological process (BP) were checked for meaningfulness. Terms unrelated to the context of the tissue investigated, such as terms related to heart or psychiatric diseases, were removed.

RNA hybridization using nCounter

Samples from the same RNA pools that were used for the QuantSeq 3′ method (150 to 250 ng) were also hybridized to the nCounter Canine IO Panel XT CodeSets (NanoString Technologies, Inc., Seattle, WA, USA), including 780 pre-selected plus 20 housekeeper probes specific for the respective genes of interest. Hybridized samples were processed following the manufacturer’s protocols (protocol ID: MAN-10023-11, MAN-10056-06) and loaded onto the nCounter MAX Analysis System’s (NanoString, Seattle, WA, USA) Prep Station for purification and immobilization on sample cartridges. These were transferred to the Digital Analyzer for data collection following the manufacturer’s user manual (protocol ID: MAN-C0035-08).

Following the workflow described in the manufacturer’s user manuals (MAN-C0019-08, MAN-C0011-04), the reporter library files (RLF) and reporter code count (RCC) files were imported into the nSolver 4.0 Analysis Software (NanoString Technologies, Seattle, WA, USA). Quality control and normalization followed default settings. Differential gene expression (DGE) analysis was achieved using the R 3.3.2-based nCounter Advanced Analysis 2.0 plug-in (version 2.0.134) with the recommended statistical settings.

In situ hybridization

In situ hybridization was performed on 4 µm FFPE sections from CCH stages 1, 2, and 3 (n = 3) and the same number of HS. The RNAscope (Wang et al., 2012) 2.5 HD Assay-RED kit (Bio-techne, Minneapolis, MN, USA) was used according to the manufacturer’s instructions (document number 322452-USM and 322360-USM). Briefly, after deparaffinization and standard pre-treatment with 1× Target Retrieval solution and RNAscope Protease Plus solution tumor sections were hybridized with target probes. A custom designed probe targeting the sequence segment 281-1239 of canine CD80 (NM_001003147.1) and a probe targeting mRNA of canine CD86 (Cat No. 578991) were used. A probe targeting bacterial dihydrodipicolinate reductase (DapB) (Cat No. 320871) was used as a negative control and tumor-adjacent tissue composed of epidermis, apocrine glands and hair follicles served as negative controls in the detection of CD80 and CD86. Additionally, a custom designed probe targeting nucleotides 2-927 of canine ornithine decarboxylase antizyme (OAZ1) (NM_001127234.1), a widely expressed housekeeper gene (de Jonge et al., 2007), was utilized as a technical positive control for the principle detectability of RNA in all cells of the respective samples (see Fig. S2). Pre-amplifiers and amplifiers were added to the samples, followed by the addition of chromogenic substrates and counter staining with Mayer’s hematoxylin (Bio-Rad, Hercules, CA, USA).

Counting of CD80 and CD86-positive tumor cells

Slides subjected to in situ hybridization were digitized at 400× magnification (0.25 µm/pixel) using an Aperio CS2 Scanner (Leica Biosystems, Buffalo Grove, IL, USA). On each section, 10 high power fields (HPF) of 220 × 120 µm were randomly selected for manual counting of all tumor cells vs signal-containing tumor cells in the respective area. In addition to the assessment of stage-specific expression in CCH, the spatial distribution of the target mRNA was measured in three equally divided horizontal tumor layers (bottom third, central third, and top third) with 3 to 4 HPF counted in each layer. To control for the principle detectability of mRNA in all cells, all tissues were additionally analyzed for the cellular expression of the housekeeper OAZ1.

MALDI-MSI

Serial sections from CCH stages 1, 2, and 3 (n = 3) were cut at 6 µm thickness, mounted in randomized order onto conductive glass slides (Bruker Daltonik GmbH, Bremen, Germany) coated with poly-l-lysine (0.05% in MilliQ-water; Sigma Aldrich, St. Louis, MO, USA) and preheated for 1 h at 60 °C. Subsequently deparaffinization was conducted by treatment with xylene followed by processing through 100% isopropanol and successive hydration steps, as previously described (Cornett et al., 2007; Hempel et al., 2023). Sections were fully rehydrated in ultrapure water (GenPurexCADPlus System; Thermo Fisher Scientific, Waltham, MA, USA). Utilizing a steamer heat-induced antigen retrieval was performed in ultrapure water for 20 min. Slides were dried for 10 min. Afterwards, tryptic digestion was performed using an automated spraying system (HTX TM-Sprayer; HTX Technologies LLC, ERC GmbH, Buchholz in der Nordheide, Germany) to deliver 16 layers of tryptic solution (20 µg Promega Sequencing Grade Modified Porcine Trypsin in 800 µL digestion buffer including 20 mM ammonium bicarbonate with 0.01% glycerol) onto each section at 30 °C. Tissue sections were incubated for 2 h at 50 °C in a humidity chamber saturated with potassium sulfate solution. Finally, four layers of the matrix solution (7 g/L α-cyano-4-hydroxycinnamic acid in 70% acetonitrile and 1% trifluoroacetic acid) were applied at 75 °C using the automated spraying system. MALDI imaging was conducted using the ultrafleXtreme MALDI-ToF/ToF device (Bruker Daltonik GmbH, Bremen, Germany) in reflector mode with a detection range of m/z 600–3,200, 500 laser shots per spot, 1.25 GS/s sampling rate and raster width of 50 μm, while FlexImaging 5.0 and flexControl 3.4 software (Bruker Daltonik GmbH, Bremen, Germany) coordinated the MALDI imaging processing. External calibration was performed using a peptide calibration standard (Bruker Daltonik GmbH, Bremen, Germany). After matrix was removed from tissue sections with 70% ethanol and tissue sections were HE-stained for histological annotation. Regions of interest (ROI) were digitally annotated utilizing QuPath software (Bankhead et al., 2017) (Version 0.2.3; University of Edinburgh, Edinburgh, UK) and transferred into SCiLS Lab software (Version 2023a Pro; Bruker Daltonik GmbH, Bremen, Germany).

MALDI-MSI raw data were imported into the SCiLS Lab software (Version 2022a Pro; Bruker Daltonik GmbH, Bremen, Germany) using settings preserving the total ion count, baseline removal, and converted into the SCiLS base data (.sbd) and simulation model (.slx) file. Peak finding and alignment were conducted across a selected dataset (interval width = 0.2 Da) using a standard segmentation pipeline in maximal interval processing mode with total ion count (TIC) normalization and weak denoising.

Protein identification by electrospray ionization tandem mass spectrometry

Protein identification for m/z values was performed on adjacent tissue sections of all three CCH stages in duplicates using a bottom-up nano-liquid chromatography electrospray ionization tandem mass spectrometry (nLC-MS/MS) approach as previously described (Hempel et al., 2023). Similar to their preparation for MALDI-MSI, sections were preheated to 60 °C for 1 h before deparaffinization. Paraffin removal, antigen retrieval and tryptic digestion were carried out as described above. Slides were incubated at 50 °C in a humidity chamber saturated with potassium sulfate solution for 2 h. Subsequently, peptides were extracted from tumor cell-rich regions separately from each tissue section into 40 μL of 0.1% trifluoroacetic acid (TFA) and incubated for 15 min at room temperature. Using a ZipTip C18 (Thermo Fisher Scientific Inc., Waltham, MA, USA) digests were filtered following the manufacturer’s instructions. Eluates were vacuum concentrated (Eppendorf Concentrator 5301; Eppendorf AG, Hamburg, Germany) and reconstituted separately in 20 µL 0.1% TFA and 4% acetonitrile (ACN). From this solution, 2 µL were injected into a nano-liquid chromatography (nHPLC) system (Dionex UltiMate 3000; Thermo Fisher Scientific Inc., Waltham, MA, USA) coupled to an Q Exactive ultrahigh-resolution mass spectrometer (Thermo Fisher Scientific Inc., Waltham, MA, USA). In order to concentrate the peptide mixture, an Acclaim PepMap100 C18 trap column (3 µm, 100 Å, 75 µm inner diameter; Thermo Fisher Scientific Inc., Waltham, MA, USA) was utilized. Subsequently, at an eluent flow rate of 300 nL/min, the peptide mixture was fractionated on an Acclaim PepMap100 C18 capillary column (2 µm, 100 Å, 75 µm inner diameter; Thermo Fisher Scientific Inc., Waltham, MA, USA). Mobile phase A comprised 0.1% (v/v) formic acid in water, while mobile phase B consisted of 0.1% (v/v) formic acid and 80% (v/v) ACN in water. Pre-equilibration of the column was conducted with 5% mobile phase B, followed by an increase to 44% mobile phase B over 100 min. Utilizing a single MS survey scan (m/z 350–1,650) with a resolution of 60,000, and MS/MS scans of the 15 most intense precursor ions with a resolution of 15,000, mass spectra were acquired in a data-dependent mode. The dynamic exclusion time was set to 20 s and the automatic gain control was set to 3 × 106 and 1 × 105 for MS and MS/MS scans, respectively.

Acquired nLC-MS/MS spectra as mascot generic files (.mgf) were matched to the UniProt reference proteome (Taxon ID: 9615, Canis lupus familiaris) using the Mascot Server (version 2.7.0; MatrixScience Inc., Boston, MA, USA). Generic settings were set to a significance threshold of p < 0.05 and the settings for trypsin as the proteolytic enzyme; a maximum of 1 missed cleavage; 0.2 peptide tolerance; peptide charges of >2+; oxidation allowed as variable modification; 0.5 Da MS/MS tolerance to identify the corresponding protein. To match aligned m/z values from MALDI-MSI with the peptides identified by nLC-MS/MS, we developed an in-house script with parameter settings as previously described (Hempel et al., 2023). Briefly, the comparison of MALDI-MSI and nLC−MS/MS m/z values required the identification of >1 peptide(s) (search mass window ≤ 0.5 Da) (Cillero-Pastor & Heeren, 2014). The peptide with the highest logP score and smallest mass difference between MALDI-MSI and nLC-MS/MS data were accepted as the correct match.

Statistical analysis

The statistical power of the QuantSeq 3′ experimental design, calculated in RNASeqPower is 0.80. The following values were used for the required parameters: Sequencing depth = 116, coefficient of variation = 0.38, effect = 2 and alpha = 0.05. Five biological and no technical replicates were utilized.

For statistical analyses, GraphPad Prism Version 9.5.1 (GraphPad Software, La Jolla, CA, USA) was used. Statistical tests applied for the different analyses are mentioned in the results section. P values of ≤0.05 were considered significant.

Results

Only minor differences between the compared groups on the mRNA level

In order to verify the initial histologic diagnosis of CCH for the selected samples, expression levels of a panel of markers for canine round cell tumors (Paździor-Czapula et al., 2015) were examined in the transcriptome data of the QuantSeq3′ analysis (see Fig. S1). The results clearly showed that the markers for CCH were significantly overexpressed in all samples compared to markers of relevant alternative round cell tumors, thus confirming the initial diagnosis of CCH for our entire study cohort.

To identify differentially expressed genes (DEGs) between CCH tumor-cell rich sample groups 1 and 2a, and between the lymphocyte-enriched areas of sample groups 2b and 3 (Fig. 1A), the QuantSeq 3′ method was applied. An average of 5,626,401 to 7,852,354 raw reads with an average input length of 66.21 to 71.65 nt was found, which resulted in 2,334,311.03 to 2,801,964.93 normalized reads per sample. Principle component analysis (PCA) was used to examine whether the compared CCH sample groups clustered or differed from each other. A strong association was found between groups 1 and 2a and likewise between 2b and 3, indicating minor differences between groups. However, a discrimination along PC1 between tumor-cell rich groups 1 and 2a vs lymphocyte groups 2b and 3 was observed (Fig. 1B). The eigenvalues of the first two principal components accounted for 60% of the total variance (PC1 45%; PC2 15%).

Figure 1 Stage-dependent transcriptome analysis of tumor cell- and lymphocyte-enriched canine cutaneous histiocytoma (CCH) groups.

(A) Schematic depiction of comparisons between different stages and areas of CCH. Areas highly enriched for stage 1 tumor cells were compared with those of stage 2, here termed group 2a, while areas highly enriched for stage 2 lymphocytes, here termed group 2b, were compared with stage 3 lymphocytes. (B) Principle component analysis plot of all compared groups in the QuantSeq 3′ analysis: Tumor cells of group 1 (red) and 2a (green) and lymphocytes of group 2b (blue) and 3 (violet). (C) Volcano plots of differentially expressed genes of group 2 tumor cells (2a) compared to group 1 tumor cells and group 3 lymphocytes compared to group 2 lymphocytes (2b). Thresholds for significantly differentially expressed genes (log2 (fold change (FC)) ≤ −1 or ≥ 1 and adjusted p-value < 0.05 ≙ −log10 (padj) > 1.3) are depicted by dashed lines.

Differential gene expression analysis revealed 249 DEGs in sample group 2a compared to sample group 1, including 58 genes with lower expression and 191 with higher expression levels (Fig. 1C). A list of all DEGs can be found in the Supplemental Material (Table S2). Between sample groups 3 and 2b, 16 DEGs were found, nine of which with lower and seven with higher expression (Fig. 1D).

For pathway enrichment analysis, g:GOSt was employed. Analyzing the down-regulated genes in sample group 2a compared to sample group 1, 16 GO terms were over-represented. However, with the exception of wound healing (−log10(p-adj) = 5.2) and response to wounding (−log10(p-adj) = 4.6), detected terms displayed adjusted p-values above, albeit very close to the threshold of significance (>0.05). This also applied to all 26 GO terms that were over-represented in the upregulated genes in group 2a compared to group 1, indicating little overall difference between the tumor cells of the two groups. A list of all biologically comprehensible and significant GO terms is presented in Table 1. No significantly enriched GO terms were found for the DEGs between sample groups 3 and 2b.

Table 1 Pathway analysis of significantly differentially expressed genes between groups 2a and 1 using g:Profiler.

Downregulated in 2a vs 1	Upregulated in 2a vs 1	
Term name	Term ID	−log10(padj)	Term name	Term ID	−log10(padj)	
Wound healing	GO:0042060	5.19	Apoptotic signaling pathway	GO:0097190	1.68	
Response to wounding	GO:0009611	4.60	Positive regulation of response to stimulus	GO:0048584	1.58	
Virion attachment to host cell	GO:0019062	1.86	Regulation of apoptotic process	GO:0042981	1.38	
Adhesion of symbiont to host cell	GO:0044650	1.81	Apoptotic process	GO:0006915	1.36	
Cell-substrate adhesion	GO:0031589	1.73	Regulation of response to stimulus	GO:0048583	1.36	
Regulation of connective tissue replacement involved in inflammatory response wound healing	GO:1904596	1.66	Regulation of programmed cell death	GO:0043067	1.36	
Regulation of connective tissue replacement	GO:1905203	1.41	Programmed cell death	GO:0012501	1.34	
Wound healing involved in inflammatory response	GO:0002246	1.39	Immune response	GO:0006955	1.34	
Developmental process	GO:0032502	1.36	Antigen processing and presentation of endogenous peptide antigen	GO:0002483	1.34	
Endothelial cell migration	GO:0043542	1.36	Immune system process	GO:0002376	1.34	
Anatomical structure formation involved in morphogenesis	GO:0048646	1.36	Regulation of monocyte chemotaxis	GO:0090025	1.34	
Extracellular matrix organization	GO:0030198	1.36	Positive regulation of monocyte chemotaxis	GO:0090026	1.34	
Inflammatory response to wounding	GO:0090594	1.36	Extrinsic apoptotic signaling pathway	GO:0097191	1.34	
Cell migration	GO:0016477	1.36	Positive regulation of chemotaxis	GO:0050921	1.34	
Cell growth	GO:0016049	1.36	Positive regulation of response to external stimulus	GO:0032103	1.34	
Regulation of cell growth	GO:0001558	1.36	Regulation of cell death	GO:0010941	1.34	
Response to bacterium	GO:0009617	1.34	
Cell surface receptor signaling pathway	GO:0007166	1.34	
Negative regulation of blood vessel morphogenesis	GO:2000181	1.33	
Antigen processing and presentation of endogenous antigen	GO:0019883	1.33	
Negative regulation of vasculature development	GO:1901343	1.33	
Regulation of apoptotic signaling pathway	GO:2001233	1.33	
Negative regulation of angiogenesis	GO:0016525	1.33	
Cell death	GO:0008219	1.33	
Regulation of cell population proliferation	GO:0042127	1.31	
Somatic diversification of immune receptors	GO:0002200	1.31	

To gain deeper insight into explicitly immuno-oncological mechanisms, the same samples used for the QuantSeq 3′ analysis were subjected to NanoString’s nCounter Canine IO Panel. Of the 800 genes covered on the panel, 790 were detected in our RNA isolates. However, none of them were significantly differentially expressed between sample group 2a vs 1 or between group 3 vs 2b. This result also includes that there was no clustering of the compared experimental sample groups for any of the gene sets included in the panel, including apoptosis, proliferation, and hypoxia. A list of all investigated gene sets is presented in Table S3.

Only minor differences between compared groups on protein level

Following a hypothesis-generating approach, we analyzed abundantly expressed peptides on the spatially resolved proteome level using MALDI-MSI technology. Highly expressed peptides in CCH stages 1, 2, and 3 were quantified and bioinformatic data processing was used to calculate which peptides were discriminating (receiver operating characteristic (ROC) value ≤0.3 or ≥0.7) between annotated areas of the different tumor stages. Similar to our transcriptome analyses, tumor cell-enriched and lymphocyte-enriched areas were compared within the different stages, respectively. A total of 412 mass features were identified across all stages (for details, see Table S4). Then, identified peptides were assigned to proteins and applied to pathway enrichment analysis with g:GOSt (Table 2). Among the calculated GO terms none was associated with established immune-oncological mechanisms, supporting the results of our transcriptome analysis.

Table 2 Pathway analysis of specifically expressed peptides in sample groups of canine cutaneous histiocytoma.

Tumor cells	
Term name	Term ID	−log10(padj)	Term name	Term ID	−log10(padj)	
Specific in 1 vs 2	Specific in 2 vs 1	
Proton-transporting two-sector ATPase complex, catalytic domain	GO:0033178	1.89	Actin filament-based process	GO:0030029	2.42	
			F-actin capping protein complex	GO:0008290	2.36	
			Negative regulation of actin filament depolymerization	GO:0030835	2.14	
			Regulation of actin filament depolymerization	GO:0030834	1.78	
			Endosome membrane	GO:0010008	1.76	
			Actin cytoskeleton organization	GO:0030036	1.55	
			Negative regulation of protein depolymerization	GO:1901880	1.50	
			Intracellular non-membrane-bounded organelle	GO:0043232	1.45	
			Non-membrane-bounded organelle	GO:0043228	1.45	
			Actin filament depolymerization	GO:0030042	1.41	
Specific in 1 vs 3	Specific in 3 vs 1	
None	Acetyl-CoA biosynthetic process	GO:0006085	1.74	
	Thioester biosynthetic process	GO:0035384	1.53	
	Acyl-CoA biosynthetic process	GO:0071616	1.53	
	Structural constituent of cytoskeleton	GO:0005200	1.43	
Specific in 2 vs 3	Specific in 3 vs 2	
F-actin capping protein complex	GO:0008290	2.57	Proton-transporting two-sector ATPase complex, catalytic domain	GO:0033178	3.12	
Negative regulation of actin filament depolymerization	GO:0030835	2.32	ATPase complex	GO:1904949	1.34	
Regulation of actin filament depolymerization	GO:0030834	1.96				
Negative regulation of protein depolymerization	GO:1901880	1.68				
Actin filament depolymerization	GO:0030042	1.59				
Cytoplasm	GO:0005737	1.48				
Regulation of protein depolymerization	GO:1901879	1.45				
Protein-containing complex disassembly	GO:0032984	1.43				
Negative regulation of protein-containing complex disassembly	GO:0043242	1.43				
Lymphocytes	
Specific in 2 vs 3	Specific in 3 vs 2	
F-actin capping protein complex	GO:0008290	2.27	None	
Cell cortex	GO:0005938	2.14		
Cortical cytoskeleton	GO:0030863	1.98		
Poly(A) binding	GO:0008143	1.55		
mRNA binding	GO:0003729	1.54		

Components of MHC I and MHC II are expressed in tumor cell-enriched groups of CCH on the mRNA and protein levels

After only minor differences were found at the mRNA and protein levels by stage comparison, a hypothesis-driven approach was applied to test for specific molecules that had previously been speculated by others (Baines et al., 2007; Kipar et al., 1998; Pires et al., 2013b) to be relevant in the immunological response that ultimately results in tumor cell destruction. This approach aimed to not only reinforce existing hypotheses but also to assess the reliability of our data.

Using the normalized counts from the whole transcriptome analysis, we specifically examined whether MHC I and MHC II complexes were expressed in the tumor cell-rich sample groups 1 and 2a. Indeed, all annotated components of both complexes except for dog leucocyte antigen (DLA)-12 were detected in both stages in substantial amounts (Fig. 2). A comparison of the number of counts of the different canine class I genes revealed that DLA-88 was highly expressed (group 1: 12,521.6; group 2a: 11,727.6), while DLA-64 (group 1: 261.5; group 2a: 165.2) and DLA-79 (group 1: 52.3; group 2a: 10.9) exhibited relatively low levels of expression.

Figure 2 mRNA expression levels of dog leucocyte antigens (DLA) major histocompatibility complex (MHC) I (A) and MHC II (B) components in tumor cell-enriched areas of select stages of canine cutaneous histiocytoma.

Means and standard deviation (n = 5) of normalized counts are depicted as detected by QuantSeq 3′ analysis.

On the other hand, the comparison of the counts of the canine class II genes revealed relatively high average expression levels of DLA-DQB1 (group 1: 1,432.2; group 2a: 1,537.4), -DQA1 (group 1: 1,259.3; group 2a: 790.5) and -DMA (group 1: 707,3; group 2a: 873.3) whereas mean expression levels of the remaining genes were relatively low, with DLA-DRA (group 1: 130.1; group 2a: 44.2), -DMB (group 1: 15.9; group 2a: 4.5), -DOA (group 1: 22.1; group 2a: 17.6) and DOB (group 1: 12,8; group 2a: 12,1).

Among the most abundant peptides as identified by MALDI-MSI, components of MHC I and MHC II were found in the tumor cell-enriched areas in all three tumor stages. MHC I peptide TFKETAQVYR was detected among discriminating peptides in stage 1 tumor cells (rank 18, ROC = 0.226) compared to stage 2. The same peptide was found to be discriminating for stage 3 (rank 1, ROC: 0.745) compared to stage 1 tumor cells, indicating that MHC I was most abundant in stage 3, followed by stage 1 and finally stage 2. This peptide sequence was found to be part of dog leucocyte antigen-88 (DLA88) which is consistent with one of the four components of the canine MHC I complex.

Similar results were found for MHC II-peptide SFDPQGALR, which was discriminating in stage 1 tumor cells (rank 7, ROC: 0.195) compared to stages 2 and 3 (rank 63, ROC: 0.76), indicating higher expression of MHC II in tumor cell-enriched areas of stages 1 compared to 2 and 3. This peptide is part of the DLA class II antigen.

Higher expression of co-stimulatory molecules than co-inhibitory molecules in CCH tumor cell-enriched sample groups on the mRNA level

To test for expression of co-stimulatory molecules, data from the QuantSeq 3′ analysis were screened for B7 family ligands in each of the tumor cell-rich sample groups 1 and 2a, respectively. Second, we tested for CD28 family receptors in each of the lymphocyte-rich sample groups 2b and 3. In fact, except for PD-1 all members of both receptor/ligand groups were found expressed in all tumor sample groups studied (Fig. 3). However, we failed to identify peptides corresponding to the respective co-stimulatory molecules in the proteome data of all three CCH stages. CD86 was most highly expressed among B7 family ligands and CTLA4 among CD28 family receptors followed by CD28 suggesting that inhibitory and activating processes are occurring simultaneously.

Figure 3 mRNA expression levels of immune regulatory B7 family ligands (A) in tumor cell-enriched areas and co-stimulatory/co-inhibitory CD28 family receptors (B) in lymphocyte-enriched sample groups of select stages of canine cutaneous histiocytoma.

Means and standard deviation (n = 5) of normalized counts are depicted as detected by QuantSeq 3′ analysis. Immuno-stimulatory (+), immuno-inhibitory (−), and apoptosis-inducing (cross) ligands/receptors are tagged. Schematic depiction of functional relationships of B7 family ligands and CD28 family receptors (C) modified from Sharpe & Freeman (2002). APC, antigen-presenting cell.

Increasing expression of CD80 and constantly high expression of CD86 in tumor cells in the time course of CCH regression

Subsequently, the hypothesis that immune response-stimulating molecules are involved in CCH regression was further specified by studying spatial- and stage-dependent expression of CD80 and CD86. For this purpose, in situ hybridization was employed to examine how many of the tumor cells expressed either of the two members of B7 family ligands. To this end, all tumors were analyzed separately in three horizontal layers, including the bottom, center, and upper third of each sample. Furthermore, cellular expression of CD80 and CD86 in each of the three layers was compared between the three tumor stages, respectively. We found that both CD80 and CD86 were expressed in the majority of tumor cells, but not in the epidermis, apocrine glands or endothelial cells. A subset of lymphocytes also expressed CD80, unlike CD86, which was not detected in lymphocytes at all. Interestingly, there was a tendency towards increased CD80 expression in the middle and upper layers from stage 1 to stage 3 (Figs. 4A –4C). A significant (unpaired t-test with p = 0.0161) difference in the center between groups 1 and 3 (Fig. 4B) was observed. In contrast, the percentage of CD86 expressing tumor cells remained rather constant in all layers across the three stages, ranging between 91.1% and 98.9% expressing tumor cells (Figs. 4D–4F).

Figure 4 Spatial and time-dependent mRNA expression levels of co-stimulatory B7 family ligands CD80 (A–C) and CD86 (D–F) in selected stages of canine cutaneous histiocytoma as analyzed by in situ hybridization.

Percentage of positive cells in individual samples and means. Asterisks (*) indicate statistically significant difference between groups 1 and 3 in the center (unpaired t-test, p = 0.0161).

Lower expression of CD86 but not CD80 in HS compared to CCH tumor cells

Finally, we tested the hypothesis whether canine histiocytic sarcoma (HS), a malignant canine tumor of interstitial DC origin that does not undergo regression but usually progresses with poor prognosis, lacks CD80 and CD86 expression. This could have been one contributing factor to the malignancy of HS. However, both co-stimulatory molecules were expressed in all HS. Subsequently, we compared expression of CD80 and CD86 in CCH with that in HS., Mean percentages of CD80 expressing tumors cells were similar in both (mean values of 73.3% in CCH and 62.1% in HS; Figs. 5A–5C). In contrast, CD86 was expressed by 94.7% of CCH tumor cells, but only in 57.6% of HS tumor cells (p = 0.0004) (Figs. 5D–5F; mean value across all stages). In contrast to larger variations in CD80 expression, mean percentages of CD86 expressing tumor cells among biological replicates were remarkably close to each other in HS between 51.7% and 62.7%. To exclude technical artifacts and control for general RNA detectability in all cells, in situ hybridization sections were additionally hybridized with a probe for the ubiquitously expressed housekeeper OAZ1. Signals for OAZ1 expression were detected rather constantly in the vast majority of tumor cells across all HS tested (mean 97.5%, range: 95.5–99.4%), arguing for generally sufficient and even accessibility of mRNA in all samples tested.

Figure 5 Comparison of mRNA expression of co-stimulatory B7 family ligands CD80 and CD86 in canine cutaneous histiocytoma (CCH) compared to canine histiocytic sarcoma (HS).

CD80 (A, B) and CD86 (D, E) expression (red dots), percentage of positive tumor cells in individual samples and means (C, F) of CCH and HS, respectively. Asterisks (***) indicate statistically significant difference between CCH and HS (unpaired t-test, p = 0.0004). In situ hybridization with fast red (chromogen, red) and Mayer’s hematoxylin (blue) counterstain. Magnification: 600×, Inserts: 1,800×.

Discussion

The stereotypic course of CCH is characterized by basolateral infiltration of virtually all tumors by mainly CD8+ lymphocytes accompanied by expression of pro-inflammatory mediators (Cockerell & Slauson, 1979; Kaim et al., 2006). However, it is still unclear by which molecular pathways this immune response is triggered. Here, we applied two strategies, first a hypothesis-generating and second, a hypothesis-driven approach, the latter based on previous work (Baines et al., 2007; Pires et al., 2009, 2013b) on this tumor. To our knowledge, this is the first study to examine CCH using transcriptome analysis, an immune-oncologic mRNA panel and MALDI-MSI proteome data. Furthermore, the expression of co-stimulatory and inhibitory molecules was investigated in CCH for the first time, particularly the spatial- and time-dependent expression of CD80 and CD86.

When we compared CCH tumor stages 1 and 2 utilizing QuantSeq 3′, a total of 249 DEGs were detected (Fig. 1) that resulted in 42 significantly over-represented GO terms in the comparison of tumor cell enriched groups (Table 1). Among 16 overrepresented GO terms in stage 1 tumor cells, twelve GO terms (75.0%) indicated higher relevance of tissue growth, regeneration or replacement. Former studies have found inconsistent results regarding proliferation of CCH tumor cells. An immunohistochemical study analyzed the Ki67 proliferation marker and found no stage dependent differences (Pires et al., 2013a). In contrast, mitotic count, a light microscopic method for counting the number of mitoses within a defined tissue area, was higher in stages 1 and 2 compared to stages 3 and 4 in a different study (Belluco et al., 2020). The contradiction between these studies remained unresolved so far. Our present study seems to support the notion that stage 1 tumor cells indeed possess higher proliferative activity than later stages. Clearly, this scenario is well in agreement with the generally accepted concept of early proliferation of this tumor, followed by immunological response and regression at later stages (Moore, 2016). Four remaining GO terms (25%) indicated a higher importance of cell adhesion in stage 1 tumor cells which is in concordance with previous observations on decreased E-cadherin expression during the course of CCH regression (Pires et al., 2009). E-cadherin is a transmembrane adhesion protein of adherens junctions that attach Langerhans cells to keratinocytes in the normal epidermis (Harrington & Syrigos, 2000; Suzuki & Takeichi, 2008). Further, it is an established marker of naïve Langerhans cells (Cumberbatch, Dearman & Kimber, 1996; Schwarzenberger & Udey, 1996). Therefore, our results are consistent with the previous hypothesis according to which CCH tumor cells change their phenotype from immature to mature antigen-presenting cells over time (Baines et al., 2007; Pires et al., 2009, 2013b).

Among 26 overrepresented GO terms found upregulated in stage 2 tumor cells compared to stage 1 tumor cells, thirteen GO terms (50.0%) were attributed to immunological responses to a stimulus, in line with increasing infiltration of mainly CD8+ T cells accompanied by an increase of pro-inflammatory mediators (Kaim et al., 2006). Nine GO terms (34.6%) suggested that apoptosis is of greater relevance in stage 2 CCH tumor cells. This stands in contrast to two previous studies that failed to detect stage dependent differences in apoptotic activity (Kaim et al., 2006; Pires et al., 2013a). One report described a constantly higher percentage of apoptotic than proliferating tumor cells in all tumor stages (Pires et al., 2013a). Supporting this, only one GO term (3.8%) related to proliferation compared to nine times more apoptosis-related terms was found over-represented in stage 2 compared to stage 1 tumor cells. The remaining two terms (7.7%) were related to negative regulation of angiogenesis suggesting that angiogenesis is less relevant in stage 2 than stage 1 tumor cells. This might be explainable with the former mentioned stronger growth in stage 1. However, in one study, the angiogenesis of CCH tumor stages was investigated based on the microvascularization density and the expression of the angiogenesis markers vascular endothelial growth factor (VEGF-A) and its receptor VEGFR-2. No significant differences were found between CCH stage 1 and 2 in this study (Costa et al., 2020).

Much to our surprise and in contrast to our QuantSeq 3′ data, no significantly DEGs and thus no over-represented gene sets were detected in the immune-oncology gene panel analysis. A possible explanation may relate to the different underlying bioinformatic processing of the data. To calculate significant DEGs from the QuantSeq 3′ data, DESeq2 (Love, Huber & Anders, 2014) was used in this study, a standard technique for whole transcriptome analysis in which the Benjamini-Hochberg (BH) procedure (Benjamini & Hochberg, 1995) is utilized to adjust p-values. In contrast, the Benjamini-Yekutieli (BY) procedure (Yoav & Daniel, 2001) is recommended for calculating the significance of DEGs in the nSolver software. BY is a more conservative and discriminating method (Goeman & Solari, 2014; Yoav & Daniel, 2001) for significance testing than BH, resulting in a higher threshold of significance and thus comparatively fewer significantly DEGs are calculated. This may well explain the different outcomes of our two approaches.

Interestingly, we found all members of the B7 family ligands and CD28 family receptors except for PD-1 to be expressed in CCH (Fig. 3). Of these, receptor-ligand pairs those that initiate lymphocyte inhibition as well as lymphocyte activation and proliferation were expressed. This corresponds to the known expression patterns of CTLA4 and CD28. It is a generally accepted concept that both activating and inhibiting signals are necessary for a balanced immune response (Sharpe & Freeman, 2002) and that CTLA4 is strongly upregulated as a result of T cell activation (Linsley et al., 1996), while CD28 is constitutively expressed (Linsley et al., 1994). The lack of expression of PD-1 by lymphocytes is consistent with the successful immune response that results in the regression of CCH. Unfortunately, this was not confirmed on the protein level using MALDI-MSI data, most likely due to the fact that this technique can only detect a few hundred of the most abundant peptides, here possibly not including co-stimulatory or -inhibitory molecules. However, a recent study detected CD80 and CD86 at the protein level in CCH via immunohistochemistry (Diehl & Hansmann, 2024). The detection of both molecules in two independent studies at the RNA and protein levels lends further support to the robustness of these findings.

According to our in situ hybridization data, CD86 was constantly expressed by CCH tumor cells independent of tumor area or stage of regression, which was recently confirmed on the protein level (Diehl & Hansmann, 2024). In contrast, the number of CD80 expressing tumor cells was found to increase with time in the central and top thirds of the tumor in this study (Fig. 4) whereas a recent study showed a decrease in its expression at the protein level (Diehl & Hansmann, 2024). The fact that an absolute decrease in protein expression was found in Diehl & Hansmann (2024) while we find a relative increase on the RNA level lends itself to interesting follow up investigations. Our observation seems to be well in line with previously described changes in expression patterns and largely overlapping functions of the two molecules (McAdam, Schweitzer & Sharpe, 1998). Specifically, both are able to induce a stimulatory signal for lymphocytes via binding to CD28 and an inhibitory signal via interaction with CTLA4. However, in most APC populations CD86 is expressed constitutively and is strongly upregulated in activated cells whereas expression of CD80 is inducible de novo, typically occurring at later time points after cell activation (Carreno & Collins, 2002). Thus, expression of CD86 and later CD80 is thought to represent sequential events (Esensten et al., 2016; Freeman et al., 1993; Hathcock et al., 1994) with higher importance of CD86 in the early initiation of immune responses (Sharpe & Freeman, 2002). Assuming similar roles of CD80 and CD86 in CCH, their expression patterns as observed in our study therefore seem to strengthen the hypothesis according to which CCH tumor cells gradually adopt the phenotype of mature APCs. In particular, CD80 with its later increase seems to be a useful indicator of CCH tumor cell maturation in this scenario, whereas the consistently high expression of CD86 may be interpreted as an indicator of an onset of tumor cell maturation earlier than the clinically apparent stage 1.

Additionally, we tested the hypothesis according to which HS, a malignant tumor originating from APCs, does not express co-stimulatory molecules that could trigger activation and proliferation of cytotoxic lymphocytes. Our data revealed that the percentage of CD80-expressing HS cells was on average similar to that of CCH. In contrast, the percentage of CD86-expressing HS cells was consistently lower than in CCH (Fig. 5), possibly pointing towards weaker activity of CD86 in the more dedifferentiated and malignant tumor. This finding may contribute to the observed differences in regression, but it cannot be considered a definitive explanation. Further investigation is necessary to confirm this hypothesis.

Critical roles of co-stimulatory molecules have previously been suggested for other canine tumors and other conditions. For example, differential expression of CD80 and CD86 may correspond with disease progression and prognosis in tumor cells of hematogenic origin (Matulonis et al., 1996; Van Gool et al., 1997). Also, dysregulation of B7 pathways is involved in autoimmune diseases such as systemic lupus erythematosus and rheumatoid arthritis (Zhang & Vignali, 2016). Furthermore, a crucial role of co-stimulatory molecules has been demonstrated in CD80 and CD86 deficient mice that develop severe deficits in both the humoral and cellular immune responses (Borriello et al., 1997). Finally, because of their critical roles in immune regulation targeting, B7 ligands have also become attractive targets for novel therapies to combat cancer and autoimmune disorders, including in humans (Chen et al., 2020). Overall, our data on co-stimulatory molecules in CCH encourage future studies on their potential to trigger anti-tumor response in tumors without natural spontaneous regression.

Furthermore, as reported in previous studies, our mRNA and protein data clearly revealed that both MHC I and MHC II components were present in all stages of CCH (Fig. 2). The presence of these components was interpreted as strong evidence for the expression of MHC I and II. MHC I had so far been only detected in cultured cells of stage 1 CCHs (Baines et al., 2007). Our data now confirm these results on genuine tumor tissue and add to previous studies the detection of MHC I in stage 2 and 3 tumor cells. Among others, one mechanism of tumor cells to evade proper immunosurveillance is downregulation of MHC I (Bujak et al., 2018), which seems not to occur in CCH.

On the other hand, MHC II had previously been shown to be increasingly expressed by CCH tumor cells from the basolateral area to the epidermis (Kipar et al., 1998) and from intracytoplasmic to membranous (Pires et al., 2013b) over time. This has been interpreted as further evidence of tumor cells adopting the phenotype of mature antigen-presenting cells. Together, the abundant expression of MHC I and II as seen in this study and by others may well add to the successful targeting of the tumor by the immune system.

Conclusions

We hypothesized that differences in mRNA and/or protein expression levels between the different stages of CCH may uncover immuno-oncological mechanisms related to the stereotypic spontaneous regression of CCH. As our major finding, all methodical approaches consistently failed to identify changes in expression levels of relevant pathways, suggesting that major immuno-oncological pathways may not be regulated during the clinical course of CCH as defined by stages 1 to 3. We speculate that key processes leading to tumor regression may occur a time earlier than examined here, which should become the subject of futures studies.

Second, we addressed expression of co-stimulatory molecules and MHC complexes that may determine decisive anti-tumor immune responses. Here, our results clearly support a role of co-stimulatory molecules in CCH regression involving cytotoxic lymphocytes. Our findings further support the hypothesis according to which CCH tumor cells adopt the phenotype of mature APCs over time. Future studies will have to address this hypothesis on the functional level.

Supplemental Information

Supplemental Information 1 mRNA expression levels of round cell tumor markers according to Paździor-Czapula et al., 2015 from canine cutaneous histiocytoma groups 1, 2a, 2b and 3.

Box-and-whisker plots display values as maximum, minimum, median, lower and upper quartiles (n = 5 or 6) of normalized counts as detected by QunatSeq 3′ analysis. The mean baseMean of all detected RNAs (132, dashed line) is provided as a basis for comparison to the number of respective round cell tumor marker counts.

Supplemental Information 2 mRNA expression patterns of CD80, CD86.

Ornithine decarboxylase antizyme 1 (OAZ1) and dihydrodipicolinate reductase (DapB) in tumor cells of canine cutaneous histiocytoma (CCH) and canine histiocytic sarcoma (HS), as well as lymphocytes, epidermis, blood vessels and apocrine glands of tumor adjacent tissue of CCH. OAZ1 served as control for RNA accessibility. DapB served as a negative control. I. situ hybridization with fast red (chromogen, red) and Mayer’s hematoxylin (blue) counterstain. Magnification: 600×.

Supplemental Information 3 RNA quality number and DV200 of samples used for QuantSeq 3′ and nCounter Canine IO Panel analyses.

Supplemental Information 4 Differential expressed genes of QuantSeq 3′ analysis.

Supplemental Information 5 List of gene sets investigated using the nCounter Canine IO Panel.

Supplemental Information 6 Abundantly expressed peptides from MALDI-MSI.

Supplemental Information 7 Results log2FC 0.5.

The authors thank Hedwig Lammert at the Charité University Medicine Berlin, Tiina Berg and Magdalena Mecking from Lexogen Services, Vienna.

Additional Information and Declarations

Competing Interests

Author Contributions

Data Availability

The authors declare that they have no competing interests.

Alina K. Loriani Fard conceived and designed the experiments, performed the experiments, analyzed the data, prepared figures and/or tables, authored or reviewed drafts of the article, and approved the final draft.

Alexander Haake analyzed the data, authored or reviewed drafts of the article, and approved the final draft.

Vladimir Jovanovic analyzed the data, prepared figures and/or tables, authored or reviewed drafts of the article, and approved the final draft.

Sandro Andreotti analyzed the data, prepared figures and/or tables, authored or reviewed drafts of the article, and approved the final draft.

Michael Hummel conceived and designed the experiments, authored or reviewed drafts of the article, and approved the final draft.

Benjamin-Florian Hempel conceived and designed the experiments, analyzed the data, authored or reviewed drafts of the article, and approved the final draft.

Achim D. Gruber conceived and designed the experiments, analyzed the data, authored or reviewed drafts of the article, and approved the final draft.

The following information was supplied regarding data availability:

QuantSeq 3′ data and nCounter data are available at the Gene Expression Omnibus (GEO) Database: GSE261387 and GSE261395, respectively.

Mass spectrometry proteomics (.raw) and imaging (.imzML) data are available at the ProteomeXchange Consortium via the MassIVE partner repository under project name “Immuno-Oncologic Profiling by Stage-Dependent Transcriptome and Proteome Analyses of Spontaneously Regressing Canine Cutaneous Histiocytoma”: PXD050403.

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
