# Peer review of "Immuno-oncologic profiling by stage-dependent transcriptome and proteome analyses of spontaneously regressing canine cutaneous histiocytoma"

_PeerJ, doi:10.7717/peerj.18444_

## Round 0.1 · original submission · Minor Revisions

The experts found your study well designed and interesting. They provide suggestions to improve your manuscript. In addition Reviewer 2 recommended that you do more analysis with different settings to identify immune-related pathways and genes defining the tumor regression. Reviewer 2 for example indicated that using the suggested tools, type 1 interferon pathway was revealed.

Reviewer 1 ·

Basic reporting

1. The language is clear
2. Sufficient background and literature are provided and cited.

Experimental design

The experimental design is reasonable. However, the sample size is small.

Validity of the findings

The work is interesting. The following should be addressed
Results
1. What is the MHC I peptide TFKETAQVYR? “This peptide sequence was found to be part of dog leucocyte antigen-88 (DLA88) which is consistent with one of the four components of the canine MHC I complex”? What is significance of this finding? Does this peptide have any role in the immune response and tumor regression?
2. “Similar results were found for MHC II-peptide SFDPQGALR, which was discriminating in stage 1 tumor cells (rank 7, ROC: 0.195) compared to stages 2 and 3 (rank 63, ROC: 0.76)”. Again, what is the significance? More analysis should be performed to determine this finding in the immune response and tumor regression.

Discussion
1. Discussion should be shortened and simplified. It contains numerous descriptions which probably should belong to the Results section.

·

Basic reporting

- Unambiguous English is used throughout.
- Raw data sharing is accessible.
- The present study extensively investigated transcriptomic and proteomic changes that could define different stages of CCH with a particular focus on regression. The present research aimed to elucidate the clinical relevance of CCH progression regarding tumor and immune cell components. Novelty is accepted when it comes to experimental approaches (comparing stage-dependent transcriptomic and proteomic alteration in CCH tumor and immune cells), by applying multi-omics to histopathologic specimens. However, it sensed that authors mainly concentrated on the expression of CD86 and CD80, which was found to have already been investigated by previous studies. In the big picture, the verification of the mRNA/protein expression on the tumor cells does not translate to elucidating mechanisms by which CCH tumor cells undergo spontaneous regression. Of note, previous studies have demonstrated that initial infiltration of mainly CD4+ T cells and to some extent granulocytes, followed by CD8+ T cells as well as B cells and macrophage in the tumor microenvironmental niche, and their proinflammatory cytokines, play a critical role in the CCH regression. Inhibition of cancer immunity characterized by FOXP3 Treg infiltration and anti-inflammatory cytokines seemed not as relevant as the activation of cancer immunity during regression. Although the author’s initial beginning of this study is very interesting and could improve cancer immunological points to CCH regression, a major concern was raised by the still lack of answers for the specific immuno-oncological dynamics responsible for the regression of CCH, as well as the lack of suggestions for reasonable further studies.

Experimental design

- Scope might fit the journal. However, the section needs to be changed. This study is not related to bioinformatic analysis. Authors have dealt with genes.
- Authors need to describe how to immunohistochemically differentiate CCH from round cell tumors in the material and method section. The possibility of misdiagnoses needs to be addressed (PMID: 25563490). In addition, it is highly needed to show representative immunohistochemical results to justify their classification (e.g., lines 195, two separate, tumor-cell-rich or lymphocyte-rich areas) based on tumor and T cell separation. Otherwise, this should be a contaminating issue and limitation.

Validity of the findings

- Authors were found to apply very stringent statistical criteria to identify DEGs, such as adjusted P value < 0.05. I understand this could reduce false positivity. However, in the case of no meaningful transcriptomic alteration found and/or in this screening step, it is better to use conventional P value criteria < 0.05 (or even FC 0.5) to identify DEGs. It could also increase the power of identifying novel genes and more importantly the power of gene set enrichment analysis. Before a clear and solid conclusion is made, authors should avoid emphasizing negative results. The claim that lack of noticeable difference in the transcriptomic changes of tumor and immune cells weakened the impact and novelty of this study. There are also online user-friendly tools for gene set enrichment with canine species selected, such as https://geneontology.org/ or http://bioinformatics.sdstate.edu/go74/. When I got a chance to run this analysis using DEGs listed in Table S2, type 1 IFN signaling was significantly enriched, consistent with previous findings: the importance of the IFN signaling pathway. Gene enrichment or ontology analysis should use all DEGs rather than using either up or down DEGs because biological processes occur by both up and down DEGs.

Additional comments

- The authors tried their best to include all relevant references to make the latest comprehensive introduction. For example, a recent study (PMID: 38962703) has investigated very similar topics compared to what the authors aimed to study herein. For example, CD80 expression was negatively associated with CCH progression (PMID: 38962703), which could highlight the importance of this work. For example, lymphocyte-mediated immune responses by the previous study authors cited. For example, cell proliferation or apoptosis (PMID: 23564778). For example, cell-cell interaction and releasable cytokines have been postulated to play a role in the elimination of CCH tumor cells.
- Please refer to PMID: 16764690. In this study, all significant changes between groups were restricted to the group I CCH versus groups II, III, and IV. In other words, study design with rationale needs to be strengthened. Bulk RNA-seq or similar experimental approaches, except for single-cell RNA sequencing, can easily expect no remarkable difference when the II, III, and IV CCH groups were compared.
- The introduction could be improved by providing a review of cytokine profiling (IL-2, TNF-α, IFN-γ, and iNOS), immune checkpoint expression, and apoptotic tumor cells associated with CCH.
- lines 233: Please provide a rationale for using the old canine reference version canfam1.0. Canfam3 was demonstrated (by our group and others) to have a better transcriptomic resolution than 1.0. In addition, there were also canfam 4 and 6 available. It is strongly recommended to apply canfam3 or 4 to the current data.
- Figure 3: Why there was no PD1 gene detection? There was no discussion of this issue.
- The rationale for including and comparing CCH with histiocytic sarcoma is weak. Not sure if this comparison is mandatory or could improve the current value of the study. In addition, currently, the dendritic tumor cell concept is debatable (Indeed, in the area of a non-tumor-existing immune environment, the tumor shows regression based on immune infiltrates that consisted of various immune subsets, such as DC, CD8+ T, CD4, B, etc). It could be only applied to tumor cells showing regression, whereas histiocytic sarcoma is never regressed based on experience. Likewise, if CCH and histiocytic sarcoma express antigen-presenting molecules, why histiocytic sarcoma is not regressed? That is, a story using CD80 and CD86 cannot be strongly explainable and thus should be avoided.
- Please include the recent study that shows the opposite results to what authors found regarding the expression pattern of CD80 on tumor cells (PMID: 38962703). It is reported that while regression, CCH tumor cells reduce the activation phenotype of DC by the loss of CD80.
- Lines 62: Please remove seemingly. It is inappropriate in the abstract.
- Lines 63-65: Inappropriate statements for abstract. Please make it more objective.
- LInes 65-66: I do not think that current results support the role of the molecular in CCH regression.
- Lines 84-93: Please include references associated with cytokine alterations such as IFN, iNOS, IL2, and others to support the role of key immune players, such as cd8+ T cells.
- lind 113: add cytokine as a third signaling for T cell proliferation
- Lines 114: Please describe the third role of T cell activation: cytokine.
- lines 526-527: There is no strong rationale for why authors investigate inhibitory immune checkpoint genes, given that previous studies have indicated that (1) PD-L1 and CD86 are not related to regression of CCH, (2) CCH is not exclusively associated with the inhibitory immune environment due to lack of inhibitory cytokine changes (TGFB, IL10) and regulatory T cell response (PMID: 16764690, 32783525, 38962703)
- There was no protein name but the description is available for mass spec results. Please provide the protein name (Supplementary Data 4).
- Authors should try to identify immune-related pathways and genes to achieve their aim based on new analyses suggested above, given that CCH is an obviously immunologically activated tumor. In addition, the authors should provide examples of how the findings in this study could be correlated with comparative oncology and translated to human cancers, which is lacking now.

Reviewer 3 ·

Basic reporting

Canine cutaneous histiocytoma (CCH) originates from dermal Langerhans cells and primarily affects young dogs. Previous studies have suggested that anti-tumor immune responses may contribute to the spontaneous regression of CCH, but the details remain speculative.
This study aimed to elucidate the specific immuno-oncological dynamics underlying CCH regression at the mRNA and protein levels. Techniques used included QuantSeq 3’ mRNA sequencing, nCounter RNA hybridization, and MALDI-MSI on formalin-fixed, paraffin-embedded CCH samples at different tumor stages.
The results indicated only minor stage-specific differences in the molecular analysis. Furthermore, major immuno-oncological pathways were not found to be regulated during CCH regression at the mRNA or protein levels by the methods employed. However, the findings may support the involvement of immune-stimulatory B7 family ligands and CD28 family receptors in CCH regression. The authors speculate that key signals for tumor regression may occur at an interactome level or earlier than the time points examined.

Experimental design

In general, this study was conducted with rational experimental designs and presents interesting findings.

Validity of the findings

However, several issues should be addressed to enhance the comprehensiveness and impact of the results:
1. Results: For lines 423, 424, 445, 446, 447, and 452, I suggest organizing the data on ranking and ROC for discriminating stages into a table to improve clarity and comprehensiveness.
2. Line 432: Correct the typo "MCH II" to "MHC II."
3. Rationale for MHC Detection: The rationale for detecting and comparing MHC I and MHC II levels rather than localizations between groups or stages is unclear. Please elaborate on the reasoning behind this analysis.
4. Lines 459-466: Given the heterogeneity of CCH tissues, it may not be appropriate to conclude that the mRNA levels of CTLA4 and CD28 are increased between groups. This should be reconsidered or further clarified.

Additional comments

1. Introduction: When describing the pairs of costimulatory or inhibitory molecules expressed on APCs and T cells (lines 107-124), it would be clearer to discuss them separately in terms of expression by cell type and their activation or inhibition when binding to their pairs.
2. Discussion: The discussion could be more concise and focused on validating the role of co-stimulatory molecules in CCH regression involving cytotoxic lymphocytes.
3. Conclusions: The statement "Future studies will have to address this hypothesis on the functional level with further elucidation of the therapeutic potential of activating co-stimulatory pathways in malignancies that do not undergo spontaneous regression" appears overstated and could be refined for clarity. Consider revising this to improve grammatical accuracy and precision.

---

## Round 0.2 · accepted · Accept

All the concerns have been addressed. One reviewer and the editor have evaluated that the revised manuscript is acceptable in the present form. Therefore, the manuscript is ready for publication.

·

Basic reporting

The authors have taken care of my concerns regarding the manuscript. The revised manuscript is believed to expand immunological knowledge of CCH.

Experimental design

Experimental design was appropriate.

Validity of the findings

Experimental validation was performed in an appropriate manner.